# Artificial Intelligence-Driven Design of Antisense Oligonucleotides for Precision Medicine in Neuromuscular Disorders

**DOI:** 10.3390/genes16121468

**Published:** 2025-12-08

**Authors:** Jamie Leckie, Sunny Wu, Terryanne Standell, Toshifumi Yokota

**Affiliations:** 1Department of Medical Genetics, Faculty of Medicine and Dentistry, University of Alberta, Edmonton, AB T6G 2H7, Canadajunyao10@ualberta.ca (S.W.);; 2The Friends of Garrett Cumming Research & Muscular Dystrophy Canada HM Toupin Neurological Sciences Research, Edmonton, AB T6G 2H7, Canada

**Keywords:** antisense oligonucleotide (ASO), artificial intelligence (AI), machine learning, neuromuscular disease, design

## Abstract

Rare neuromuscular disorders impose a significant burden on patients, caregivers, and the health care system, yet, effective disease-modifying therapies remain limited. Antisense oligonucleotides (ASOs) have emerged as a promising therapeutic strategy, enabling targeted modulation of gene expression through mechanisms such as exon skipping, exon inclusion, and transcript degradation. However, the clinical efficacy of currently approved ASO therapies is often suboptimal. This limitation reflects not only poor target tissue uptake and delivery barriers, but also suboptimal design of ASO sequences and chemical modification patterns, which can compromise potency, safety, and translational robustness. Recent advances in machine learning have led to the development of ASO optimization platforms such as eSkipFinder and ASOptimizer, which aim to predict effective ASO sequences and chemistries for specific RNA targets. While these tools show considerable promise, their broader applicability remains limited due to a lack of comprehensive validation and the absence of integrated safety considerations. Further refinement and validation are necessary to improve their translational utility. Nevertheless, such platforms represent a critical advancement toward accelerating ASO development. By improving design precision, reducing reliance on extensive preclinical screening, and enabling researchers with limited ASO experience to generate optimized candidates, machine learning is poised to accelerate the development and clinical translation of ASO therapies for rare neuromuscular disorders.

## 1. Introduction

Rare neuromuscular disorders pose a substantial burden on both healthcare systems and affected individuals, often leading to significantly reduced life expectancy and diminished quality of life for patients and their caregivers [1,2,3,4]. Although each disorder affects a small proportion of the population, typically fewer than 1 in 100,000 individuals [5], collectively, rare neuromuscular disorders are estimated to impact approximately 15 million people globally [6]. Clinical presentations and disease severity vary widely depending on the underlying genetic cause. However, most patients experience progressive muscle weakness and impaired mobility, resulting in substantial difficulty performing daily activities [7]. In many cases, these disorders are further complicated by life-threatening respiratory abnormalities, cardiac involvement, and dysphagia [8,9].

Despite the significant clinical and social burden of these chronic, often progressive disorders, fewer than 5% of rare diseases currently have FDA-approved treatments [10]. The rarity and genetic specificity of each condition present substantial therapeutic challenges as most rare neuromuscular disorders require novel therapeutics, and in some cases, treatment may only be applicable to a subset of patients with specific pathogenic variants [11]. A promising and versatile approach to treating rare neuromuscular disorders is antisense oligonucleotides (ASOs), which can regulate RNA through multiple mechanisms to treat disease [12]. ASOs have been FDA-approved for numerous rare neuromuscular disorders, with many ongoing clinical trials for additional diseases currently underway [13]. Although these ASOs are often observed to result in improvements in patients’ symptoms, significant room for improvement remains for ASOs, largely because they exhibit only substandard delivery to target cells and successfully through the cell’s membrane [14,15]. Numerous approaches to improve overall ASO efficacy have been investigated, with clinical trials underway for numerous delivery systems and ASO conjugates [16].

An alternative approach to optimizing ASO efficacy is to improve the overall effect through the sequence and pattern of chemical modifications used. Since the FDA approval of ASOs for rare neuromuscular disorders, subsequent studies have reported optimized ASOs with improved sequence design and/or chemical modifications, demonstrating greater efficacy than those currently approved [17]. Historically, tools to identify optimal ASO sequences could only take into consideration one or two ASO features [18,19]. Due to the complexity of ASO-mRNA binding differences between different ASOs and potential mRNA targets, as well as the vast range of potential sequences, artificial intelligence (AI) is a powerful tool that has allowed researchers to predict optimized ASO sequences and/or chemical modification patterns with machine-learning predictive models trained on the vast amount of experimental data available on ASOs. These platforms, including eSkipFinder and ASOptimizer, have significant potential to reduce time and costs associated with preclinical trials while increasing the likelihood of identifying a more optimal sequence for improved efficacy in patients [18,20]. This review provides a comprehensive overview of ASOs and their design, the mechanisms they can utilize to treat rare neuromuscular disorders, and AI-driven platforms that have promising capabilities to advance their translation to a clinical setting to benefit patients with rare genetic neuromuscular diseases.

## 2. Overview of Antisense Oligonucleotide Mechanisms and Design

ASOs are small DNA-like molecules, on average 12–25 nucleotides in length, that are capable of Watson-Crick base pairing to mRNA containing their complementary sequences [12,21]. Once bound to mRNA, ASOs can modify the transcript’s splicing or cause its degradation, allowing them to modulate the expression of mutations at the RNA level to slow the progression of genetic diseases [13,22]. In theory, ASOs can be designed to be complementary to any sequence, and thus can target the expression of any RNA transcript [23]. This versatility makes them an attractive potential treatment for rare genetic neuromuscular diseases, offering a means to target specific mutations in individuals or small groups.

The therapeutic effects of ASOs are achieved through a variety of mechanisms that lead to modified mRNA splicing, RNA degradation, or steric blockage of translation machinery [24,25]. Although numerous mechanisms exist depending on the ASO design, these mechanisms are often broadly classified into two major categories: RNase H-dependent mechanisms, which cause degradation of the RNA transcript utilizing an endogenous enzyme known as RNase H1, and RNase H-independent mechanisms, which primarily modify splicing or translation through steric blockage [26,27].

### 2.1. RNase H-Dependent ASO Mechanisms

Upon binding, many ASOs depend on an enzyme known as RNase H1 to cleave target mutant RNAs [26]. RNase H1 is one of two endogenous RNase H isozymes in human cells and is responsible for recognizing and degrading DNA-RNA heteroduplexes [28]. DNA-RNA heteroduplexes are formed between single-stranded RNA and single-stranded DNA during DNA transcription, replication, and repair [29,30]. While formed during many natural processes, they can lead to genomic instability and immune activation when persistent [31]. Once bound by RNase H1, the mRNA within a heteroduplex is cleaved while the DNA is released intact [32]. Due to their DNA-like qualities, ASOs can be designed to utilize this mechanism to target mRNA with disease-causing mutations, forming a heteroduplex that is similarly degraded by RNase H1. This can occur in the nucleus or cytoplasm [33,34,35]. After internal cleavage by RNase H1, other endogenous nucleases further degrade the targeted RNA to completion [36]. The intact ASO unbinds, becoming available to bind more mRNA [37]. Importantly, to achieve their therapeutic effect, ASOs are chemically distinct from native DNA to improve stability and avoid endonuclease detection [36]. However, RNase H-dependent ASOs must still contain a stretch of at least five consecutive DNA nucleotides, which is necessary for RNase H1 recognition [27,32].

RNase H-dependent ASOs have the potential to significantly decrease the levels of target RNA within a cell [26,35]. Thus, their mechanisms are particularly useful for diseases caused by gain-of-function mutations, in which the presence of the mutant proteins is responsible for the disease phenotype, and preventing their production can improve symptoms [38]. Such mutations are the case for neuromuscular disorders, including amyotrophic lateral sclerosis (ALS; OMIM #105400), in which mutant proteins form pathological aggregates and lead to cytotoxicity [39]. Targeting the production of these mutant proteins with RNase H-dependent ASOs can improve the disease phenotype [40,41].

### 2.2. RNase H-Independent ASO Mechanisms

ASOs can also be modified in a way that prevents recognition by RNase H1, instead utilizing alternative mechanisms to modulate RNA expression. Instead of targeting mRNA for degradation, RNase H-independent mechanisms have the potential to modulate pre-mRNA splicing in a way that restores function to the mutant protein, making them good candidates for the treatment of diseases arising from loss-of-function mutations [38,42,43,44]. Although RNase H-independent ASOs often exert their therapeutic effect by modulating the splicing of mRNA, they can also be designed to target other steps in the processing of pre-mRNA, ultimately affecting translation or stability [24].

Many mutations responsible for neuromuscular disorders are loss-of-function variants that result in nonfunctional proteins. This is often the result of aberrant splicing events, which can arise from mutations to splice donor and acceptor sites or splicing regulatory elements leading to a final mRNA isoform that shifts the reading frame and may result in a premature stop codon [43,45,46]. These splicing errors can also lead to the loss of critical exons [47,48]. Structural mutations within a gene, such as deletions, duplications, and insertions, can also abolish protein function. They may create a premature stop codon by abolishing the mRNA reading frame or removing critical regions of a gene, creating a nonfunctional protein [49,50,51,52].

ASOs can modulate mRNA splicing by sterically blocking splice sites from interacting with the spliceosome, altering the combination of exons in the final transcript to create a less detrimental isoform [24,53]. The occlusion of a splice site from spliceosome access can promote the exclusion of specific exons [43]. Exon skipping is notable for its ability to restore the reading frame of an mRNA transcript that has been disrupted, such as in the case of many patients with Duchenne muscular dystrophy (DMD; OMIM #310200) [54]. DMD is a fatal muscle degenerative disorder most commonly caused by out-of-frame exon deletions or duplications in the *DMD* gene, leading to the loss of dystrophin protein [55]. For DMD, ASOs have been developed to induce the skipping of specific exons adjacent to the deletion or duplication, which restores the dystrophin reading frame [54,56]. This creates an internally deleted but functional dystrophin protein, allowing for slower disease progression and muscle degeneration [54,56]. Aside from compensating for out-of-frame deletions and duplications, exon skipping is also a promising potential approach to preventing protein truncation through the exclusion of cryptic exons or poison exons, as well as restoring function to toxic proteins by excluding exons with gain-of-function mutations [57,58,59,60]. To achieve an opposite effect, ASOs can also be designed to knock down mRNA by promoting aberrant exon skipping and generating premature stop codons, which leaves the mRNA susceptible to nonsense-mediated decay [61]. Exon skipping can also be utilized to promote specific isoforms of a protein, restoring disease-causing isoform imbalances [62,63].

Exon inclusion can also be induced to restore functional protein expression using RNase H-independent ASOs. These ASOs induce the inclusion of certain exons by sterically occluding splice silencer sites or spliceosome-inhibiting secondary RNA structures [64]. A notable example is the treatment of spinal muscular atrophy (SMA; OMIM #253300), in which ASOs are used to sterically block a mutant splice silencer from splice repressor recognition, restoring the inclusion of an exon critical for protein function [65,66]. While exon inclusion is particularly useful for promoting the inclusion of critical exons when their loss impedes protein function, such as in SMA, it can also be used for a few other purposes. For example, exon inclusion can induce the inclusion of a poison exon to decrease the expression of disease-causing mRNA transcripts [67]. Due to difficulties in identifying reliable target sites, designing ASOs that work through an exon inclusion mechanism is often more complex than those that utilize exon exclusion [68].

Although splice modulation is the most well-studied mechanism of RNase H-independent ASOs, many sites on the target mRNA can also be targeted to achieve therapeutic effects independent of splice modulation. For example, some preclinical RNase H-independent ASOs can also prevent the expression of a harmful mutant protein by inhibiting translation. Such ASOs can bind to regulatory structures in the 5′-UTR or to the start codon of a mature mRNA transcript, sterically blocking the recruitment or function of translation machinery [28,69,70]. This mechanism further broadens the therapeutic potential of ASOs by providing a reversible means of protein expression inhibition, and it may also have fewer off-target effects [71]. ASOs could also be designed to bind the 3′ end of RNA, which could interfere with mRNA stability or translational efficiency [24,28,72]. Utilizing RNase H-independent mechanism degradation allows the ASO to tolerate more extensive performance-enhancing modifications, which may not allow RNase H1 recognition [61]. ASOs can also act like agonists to upregulate the expression of certain proteins. For example, this can be achieved by interfering with negative regulators of translation, stabilizing the transcript, or inhibiting nonsense-mediated decay, all of which may improve message abundance [73,74,75,76]. Overall, ASOs are versatile tools that can be designed to target various components of mature or premature mRNA to achieve a wide range of effects on mRNA and protein levels.

### 2.3. Chemical Modifications

Unmodified DNA is susceptible to degradation within the cell due to the presence of nucleases, limiting the therapeutic efficacy of unmodified ASOs [77]. As well, unmodified DNA binds weakly to plasma proteins, and thus is very rapidly filtered by the kidneys and excreted into the urine [32,78]. As such, various chemical modifications to the backbone and sugar moieties can be introduced to allow ASOs to evade nuclease recognition, improve their stability and binding affinity, and enhance their cellular uptake [27]. ASOs can be broadly classified into three generations based on the location within the nucleotide where the modification exists [79].

In first-generation ASOs, the phosphate backbone linking each nucleotide is modified, with the non-bridging oxygen atoms in the phosphodiester bond being replaced by various chemical groups [21]. The most common modification in this group, still used today, is the phosphorothioate (PS) modification, in which the non-bridging oxygen is replaced by a sulfur group in place of phosphate. PS modifications dramatically increase nuclease resistance and plasma protein binding, prolonging the half-life of the ASOs [25,80]. As a fundamental way to improve ASO stability, they have remained foundational in ASO modifications [81]. However, biologically, PS-modified ASOs are highly toxic due to their non-specific binding to proteins [82]. As well, the synthesis of ASOs with PS backbones is typically stereo-random, generating a stereoisomer mixture that limits the pharmaceutical efficacy of the drug [83]. Thus, replacing PO bonds with PS bonds decreases the binding affinity of the ASO to its target [81]. Such drawbacks motivated the generation of ASOs with improved binding specificity and safety, in particular the pairing of phosphate modifications with changes to the sugar moiety, which can restore ASO binding affinity, leading to the second generation of ASOs [15,21].

The second generation of ASOs is characterized by modifications to the 2′ position of the ribose sugar. 2′-O-methyl (2′-OME) and 2′-O-methoxyethyl (2′-MOE) modifications introduce oxygenated groups to this 2′ position, which gives the ASO improved binding affinity and less toxicity in comparison to PS modifications alone [84,85]. Both types of modifications also improve ASO stability and prevent nuclease recognition [86]. They also improve safety, mitigating immune activation that is often triggered by certain unmodified ASOs [87,88]. Another group of second-generation ASOs is 2′-fluoro modified ASOs, which replace the 2′ position of the ribose sugar with a fluorine atom, conferring similar benefits including improved binding affinity and nuclease resistance [89]. Although sugar modifications are often designed with common goals, they are also not made equivalent. For example, 2′-MOE modifications in RNase H-dependent ASOs offer superior knockdown efficiency and nuclease resistance, with less toxicity, when compared to 2′-OME modifications [90].

A diverse group of modifications, often more extensively throughout the backbone, comprise the third generation of ASOs with further improved binding affinity, specificity, stability, and efficacy [90]. Those that make changes to the sugar structure include locked or bridged nucleic acids (LNAs/BNAs), in which the ribose takes on a rigid, or “locked” ribose conformation that boosts thermal stability, binding affinity, and nuclease resistance [91,92,93]. This locked conformation is achieved through a methylene link that creates a bridge between the 2′ oxygen and 4′ carbon of the sugar moiety, which locks the nucleotide in a rigid conformation [94]. Ethylene nucleic acids (ENAs) are a similar variation with an ethylene bridge in place of the methylene bridge [95]. The addition of a methyl group to the LNA structure forms a 2′,4′-constrained ethyl (cEt) structure, which resembles a rigid form of the 2′-MOE modification [96,97]. These modifications further reduce liver toxicity significantly without compromising the potency of the ASO [96].

Newer ASO modifications can also alter both the sugar and the phosphate groups of the ASO backbone. In phosphorodiamidate morpholine oligonucleotides (PMOs), the natural five-membered ribose is replaced with a six-membered morpholine ring and the phosphate groups are replaced by phosphorodiamidates groups [98]. Peptide nucleic acids (PNAs) also have extensively modified, charge-neutral backbones. In PNA modifications, the sugar-phosphate backbone of an unmodified ASO is replaced with a fully synthetic pseudo-peptide N-2-aminoethyl-linked backbone [99]. This grants them even more superior binding affinity than PMOs [100]. These chemical changes grant PMOs and PNAs robust nuclease resistance due to the substantial alteration in their structure from naked DNA, as well as improved safety and decreased immune activation [101,102,103,104]. However, due to their neutral charge, PMOs and PNAs have inefficient cellular entry abilities and lower binding affinity for plasma proteins, which decreases their therapeutic potential [105,106]. This may be remedied by conjugating cell-penetrating peptides to the oligonucleotide, which can improve delivery [106,107]. In addition, PMOs have received significant attention for their use in DMD treatments, as their uptake is significantly improved in DMD cells, which possess a leaky membrane that allows these small molecules to enter the cell more easily [108,109,110].

Nucleobase modifications are also an emerging group of modifications currently in preclinical development for ASO [98]. These modifications make changes to the nitrogenous bases of the ASO, commonly aiming to improve duplex stability while maintaining or improving native base pairing and hydrogen bonding strength [111]. For example, G-clamps, also known as 9-(2-aminoethoxy)-phenoxazine derivatives, are tricyclic cytosine analogues that base pair with guanine via four hydrogen bonds [112]. When substituted in place of cytosine in ASOs, the increased number of hydrogen bonds greatly increases the binding strength and melting temperature of the ASO, which helps to increase hybridization efficacy and specificity [112,113]. Bases can also be replaced with various other chemical groups, including pseudoisocytosine, amine, thione, halogen, alkyl, alkenyl, or alkynyl groups [98]. Although promising, nucleobase modifications leave room for investigation into potential off-target effects [114,115].

Modifications to ASOs must be designed with consideration of their mechanism. While the majority of the discussed modifications prevent nuclease detection of the ASO, shielding the ASO from degradation, when they are applied to the entire ASO, they prevent the ASO from utilizing RNase H1 to target mRNA for cleavage [15,116]. Thus, RNase H-dependent ASOs must contain a region of minimally modified DNA to permit RNase H1 recruitment [26]. In addition to affecting recognition, certain chemical modifications can reduce the cleavage activity of RNase H1 [117]. As well, the specific sequence within the ASO that remains unmodified, serving as the location of cleavage, also influences ASO efficiency [118]. A predominant solution for this is the development of DNA gapmers, which are chimeric RNase H-dependent ASOs that contain a central region of unmodified DNA flanked by stabilizing and performance-enhancing modifications such as 2′-OME or LNA-modified oligonucleotides [15,84]. Notably, PS modifications are an exception in that they are still recognized sufficiently and thus can be applied to the entire drug [119]. Contrastingly, because RNase H-independent ASOs do not rely on enzymatic recognition, modifications can be made more extensively and throughout the entire backbone of the ASO [120,121].

## 3. Antisense Oligonucleotides for Rare Neuromuscular Disorders

The clinical success of ASOs has been evident in the field of rare genetic neuromuscular disorders, where their ability to target the genetic causes of disease has translated into effective therapeutic advances. Although these conditions are often individually rare and heterogeneous, many share well-defined molecular causes that make them particularly amenable to RNA-targeted strategies. Over the past decade, this has led to numerous FDA approvals for ASOs that not only address previously untreatable disorders but also exemplify the potential of precision medicine for rare genetic diseases.

Although the current FDA-approved ASOs for rare neuromuscular diseases treat monogenic diseases, there is significant potential for ASOs to also be applied to non-monogenic diseases by regulating the expression of transcripts that contribute to disease pathology. For example, myasthenia gravis (MG; OMIM #254200) is an autoimmune disease typically caused by autoantibodies targeting the acetylcholine receptors (AChRs), resulting in impaired neuromuscular transmission [122]. Patients with MG exhibit elevated levels of the “read-through” acetylcholinesterase isoform (AChE-R), which degrades acetylcholine, a neurotransmitter essential for neuromuscular function [123,124]. ASOs targeting AChE-R to promote its degradation have shown efficacy in improving MG symptoms in early clinical trials [125,126]. While long-term studies are still needed to confirm safety and persisting clinical benefits, these findings demonstrate the broader therapeutic potential of ASOs for treating non-monogenic neuromuscular disorders.

### 3.1. FDA-Approved ASOs for Rare Neuromuscular Disorders

There are currently 11 ASO drugs which have been approved by the FDA, 8 of which are for neuromuscular disorders (Table 1). The first ASO to be approved was fomivirsen, which downregulated the expression of the cytomegalovirus *IE2* gene, a key factor in cytomegalovirus retinitis that is frequently observed in AIDS patients [127,128,129]. Fomivirsen was a PS-modified ASO that was delivered intravitreally and had no reported systemic distribution [128]. This drug received FDA approval in 1998 and EMA approval in 1999 [129]. However, FDA approval of fomivirsen was withdrawn in 2002 following the advent of antiretroviral HIV therapy, which had the benefit of not only preventing cytomegalovirus retinitis, but also doing so by treating the underlying HIV infection [129]. Following a 10-year lull, ASOs began receiving renewed interest, specifically for the treatment of genetic diseases, leading to the approval of mipomersen for the treatment of homozygous familial hypercholesterolemia in 2013 and 9 more drugs in the following 10 years [13].

In 2016, eteplirsen became the first ASO approved for the treatment of DMD [130]. Unlike the two previously FDA-approved ASOs, which relied on antisense interference to decrease expression of a protein, eteplirsen uses splice modulation to increase the expression of a specific isoform. Eteplirsen induces the skipping of exon 51 in dystrophin to restore the reading frame and produce a functional, truncated version of the dystrophin protein [131,132]. This truncated protein is similar to those produced with the less severe Becker muscular dystrophy and for DMD patients represents a significant improvement in dystrophin levels and symptoms [133]. Skipping exon 51 has the potential to restore the *DMD* reading frame for 10% of DMD patients, representing the largest mutation subgroup predicted to be treatable through single exon-skipping [131,134]. From 2019–2021, three novel exon-skipping PMOs ASOs, golodirsen, viltolarsen, and casimersen, received approval for the treatment of DMD [135,136,137]. Much like eteplirsen, these PMOs aim to restore functional dystrophin; however, they target alternative exons for skipping to improve the overall accessibility of these therapeutics in the DMD patient population. golodirsen and viltolarsen promote exon 53 skipping, a strategy which has the potential to treat approximately 8% of DMD patients [131,134,138]. Casimersen induces the skipping of exon 45, with applicability for another 9% of DMD patients [139].

Nusinersen was approved in 2016 for the treatment of spinal muscular atrophy and also uses splice modulation [140]. SMA patients lack functional SMN1 protein and thus rely on the nearly identical *SMN2* gene. *SMN2* gene products are typically truncated due to a point mutation in exon 7 that converts a splicing enhancer site to a splicing silencer, which prevents the inclusion of exon 7 [141,142]. Nusinersen works by sterically blocking this mutant splice silencer site, allowing for production of the full functional protein [65,143].

Inotersen, approved in 2018, was developed for the treatment of hereditary transthyretin amyloidosis (hATTR; OMIM #105210) [144,145]. HATTR occurs when a mutation in the transthyretin (*TTR*) gene disrupts the normal tetrameric interaction of transthyretin monomers, leading to aggregation [146,147]. These abnormally aggregated complexes form amyloid deposits throughout the body, leading to extensive dysfunction across multiple organ systems. Inotersen promotes the degradation of *TTR* mRNA to reduce the amount of circulating transthyretin, thereby reducing amyloid deposits [147]. In 2023, eplontersen was approved and became the first ASO approved conjugated to another molecule [148,149]. Eplontersen uses the same mechanism as inotersen; however, it is conjugated to the carbohydrate *N*-acetylgalactosamine (GalNAc) to facilitate improved uptake in hepatocyte cells, which are the primary source of circulating transthyretin [150]. This targeted delivery and improved uptake allow for a substantially lower dose of the drug to be used when compared to unconjugated equivalents.

Tofersen was approved in 2023 for the treatment of amyotrophic lateral sclerosis, specifically in patients diagnosed with mutations in the superoxide dismutase 1 (*SOD1*) gene [151]. Mutations in the *SOD1* gene affect around 2% of ALS patients and result in a toxic gain of function that is still not well understood [152]. Tofersen combats this toxic protein by reducing the concentration of SOD1 in cerebrospinal fluid through RNase H degradation [41].

Along with the drugs discussed above, there have been a handful of ASOs that have reached N-of-1 trials to treat individual patients. These N-of-1 drugs are designed specifically for patients with very rare mutations and often have a much faster development timeline to allow patients to receive the drug before succumbing to their disease. In 2019, the ASO milasen was the first personalized therapy to be used in an investigational drug for the treatment of late-infantile Batten disease (OMIM #610951) in a single patient [153]. This patient had a mutation in the *MFSD8* gene that disrupted normal splicing, which milasen was able to partially restore through splice modulation. Treatment provided some symptomatic relief to the patient and appeared to slow symptom onset prior to the patient’s death. Using the framework created through the development of milasen, another N-of-1 drug was developed and administered to individual patients. Atipeksen modulated the splicing of the *ATM* gene for the treatment of ataxia telangiectasia (OMIM #208900) and was found to reduce the severity of symptoms in the patient it was administered to [154]. The development of these drugs and their unique stories offer a potential roadmap for the development of future N-of-1 drugs, something of particular importance for rare genetic neuromuscular diseases where unique individual mutations are common.

### 3.2. Current Challenges Associated with Antisense Oligonucleotide Therapy for Rare Neuromuscular Diseases and Potential Solutions

Despite the approval of nearly a dozen drugs over the past two and a half decades, there remains significant room for improvement in their delivery, toxicity, and design of ASOs. These challenges can impede drug development and lead to reduced clinical outcomes, preventing patients from experiencing the full benefit of these drugs.

#### 3.2.1. Delivery and Uptake

A major challenge in ASO delivery is premature degradation before the drug reaches its target. With the exception of one, all FDA-approved ASOs are unconjugated and lack a dedicated delivery vehicle, leaving their stability solely up to the nucleotide chemical modifications [155]. Even upon reaching target cells, additional barriers remain. ASOs can become trapped in the endosome they are taken up in, ultimately being trafficked to lysosomes for degradation. Within cells, protein interactions can further influence ASO efficacy, facilitating beneficial processes including endosomal escape, but also hindering activity when competing with RNase H1 for binding [156,157]. Although some ASOs manage to escape the endosome and function in the cell, this typically represents less than 1% of the intracellular drug pool [158].

Systemic delivery of ASOs is also severely limited in terms of tissue specificity. The majority of systemically administered ASOs accumulate in the liver and kidneys, making delivery to other tissues considerably more challenging [159]. Most ASOs are unable to cross the blood-brain barrier, restricting the use of systemic administration when targeting the central nervous system (CNS). Although tricyclo-PS-modified ASOs have been administered in high doses to achieve limited CNS penetration, their safety profiles remain suboptimal [160]. As such, intravitreal or intrathecal administrations must currently be used when targeting the CNS. Notably, once the ASO reaches the CNS, its half-life is substantially longer compared to systemic delivery, reducing the frequency of dosing required [143]. For example, the FDA-approved drug nusinersen is delivered intrathecally and requires only three maintenance doses annually following the initial loading doses [140]. However, intrathecal injections are associated with significant risks, including postdural puncture headaches and spinal hematoma, which may negatively impact patients’ quality of life [161]. In addition, spinal abnormalities such as scoliosis, which is common among individuals with neuromuscular disorders, can further complicate intrathecal administration [141,162].

The limitations in uptake often translate into reduced clinical outcomes, as only a small fraction of the administered drug reaches the target tissues, and an even smaller proportion can function within the cell. This challenge is observed in the currently approved ASOs for DMD, all of which exhibit poor cellular uptake and limited efficacy in restoring dystrophin expression. Among these therapies, viltolarsen demonstrated the highest efficacy in a phase 2 trial, resulting in a 5.8% increase in functional dystrophin protein, whereas eteplirsen showed the lowest, with only a 0.63% increase [43]. Although these modest increases have resulted in clinical benefit, further improving ASO efficacy remains essential to achieving meaningful therapeutic outcomes [163]. A particular concern is their poor uptake in cardiac tissue [164]. In DMD, this limitation means that while ASOs have the potential to alleviate symptoms and slow skeletal muscle degeneration, they do not prevent fatal cardiomyopathy inherent to the disease [13].

#### 3.2.2. Toxicity and Off-Target Effects

The PS backbone is the major source of ASO-associated toxicity [165]. Protein interactions with the PS backbone can disrupt coagulation pathways and activate complement responses [166,167,168]. In addition, PS-modified ASOs tend to accumulate heavily in the kidneys, liver, and lymph nodes, where they can trigger inflammation and organ-specific damage [169]. Dose- and sequence-dependent thrombocytopenia has also been reported, in some cases severe enough to require hospitalization and, rarely, resulting in patient death [147,169,170,171]. One strategy to mitigate the risks is the development of ASOs that maintain efficacy at lower doses, thereby reducing toxicities associated with higher systemic exposure [169].

Other chemical modifications designed to enhance ASO stability and uptake may also introduce safety concerns. Some modifications can trigger an immune response through engagement of pattern-recognition receptors. For example, cytosine residues within CpG motifs can activate toll-like receptors (TLRs), eliciting an immune response [172]. This, however, can be avoided by methylating the C in the CpG motifs and preventing the activation of the TLRs. Additionally, in vitro screening approaches are commonly used to assess and minimize the immunogenic potential of candidate ASOs [173,174].

Off-target binding of ASOs raises substantial safety concerns, as unintended hybridization of partially complementary transcripts can result in abnormal regulation of non-target genes [165]. While in silico modeling can reduce this risk by predicting potential off-target interactions, some ASOs are capable of binding and eliciting cleavage despite the presence of significant mismatches. In vitro assays remain critical for assessing off-target activity, as they can capture unintended events that computational predictions may overlook [175].

#### 3.2.3. Limitations in Traditional Design Protocols

The design and validation of ASO is a complex and resource-intensive process, requiring considerable time and investment [176]. Numerous factors influence an ASOs ability to achieve its intended cellular effect, including binding energy, secondary structure of the target RNA, and the chemical modifications applied [155,177,178]. While traditional design processes have led to the approval of several ASO therapies, some of these have since been outperformed by optimized ones. For example, only a year after eteplirsen received FDA approval, another group identified an exon 51-skipping ASO that produced a seven-fold increase in dystrophin restoration compared with eteplirsen [17]. This highlights the risk that patients may not always receive the most effective ASO possible. Optimizing both sequence and chemical modifications during ASO development is therefore essential to maximize clinical benefit.

To address the complexity of ASO design, researchers often use computational tools to prioritize candidate sequences and improve the likelihood of their success. Early platforms, such as Sfold, enabled modeling of nucleic acid secondary structures for input sequences to aid in predicting the ASOs efficacy [19,179]. However, despite these advances, significant limitations remain. Most existing tools can only consider one or two parameters affecting ASO efficacy, thereby restricting predictive power [18]. Many are designed broadly for RNA-targeting therapeutics and also fail to account for the intended mechanism of action [20]. Furthermore, no current platform incorporates optimization of chemical modification patterns [20]. These gaps indicate the urgent need for advanced tools capable of integrating multiple determinants of ASO activity to enhance clinical translatability and therapeutic effectiveness.

#### 3.2.4. Potential Solutions to Improving ASO Efficacy for Rare Neuromuscular Disorders

Several strategies are currently being explored to enhance the efficacy of ASOs for rare neuromuscular disorders. Some approaches focus on improving delivery, using specialized systems or small-molecule conjugates that increase cellular uptake and tissue distribution. In parallel, advances in sequence design and chemical modifications aim to optimize target binding while minimizing off-target effects, thereby improving both potency and safety.

Beyond direct chemical modifications, ASOs can be conjugated to various other small molecules to enhance stability and facilitate targeted delivery. Peptide conjugates, particularly cell-penetrating peptides (CPPs) with positively charged residues, can improve transport across cell membranes [180]. Some peptides also promote endosomal escape, further enhancing intracellular delivery. Conjugation chemistry currently limits successful conjugation to neutral PMOs [181,182], but when optimized, uptake can increase dramatically. Early studies reported up to 25-fold improvements depending on the construct and the target tissue [183]. For example, a PMO conjugated to the peptide DG9 for DMD restored dystrophin levels seven times more effectively than the commercially available Eteplirsen [184]. Similarly, in SMA, a neurotensin-conjugated ASO doubled its effectiveness in mice [185].

Antibody-drug conjugates has gained popularity due to its ability to provide targeted, predictable uptake of therapeutics, often enhancing efficacy and reducing off-target toxicity [179,180,186]. Aptamer conjugates offer a comparable approach at lower cost, with minimal immunogenicity [187]. These short oligonucleotide sequences mimic antibody binding and can be synthesized concurrently with ASOs, though small molecules are still required to facilitate endosomal escape [187].

Carbohydrate conjugates have been successfully applied in FDA-approved ASO therapies. Both the ASO eplontersen and the siRNA givosiran use GalNAc conjugation [149,188]. GalNAc binds the asialoglycoprotein receptor 1 (ASGR1) on hepatocytes, enabling targeted, predictable uptake [189,190]. By facilitating both cell entry and endosomal escape, GalNAc conjugation substantially improves drug performance compared with unconjugated ASOs [191]. This strategy has also been extended to other carbohydrate-binding proteins (lectins) for broader application in enhancing cellular uptake [192,193].

Encapsulation of ASOs in delivery vehicles is another promising avenue. Lipid nanoparticles (LNPs) have attracted attention due to their proven efficacy in other therapeutics, offering improved uptake and reduced toxicity [194]. Preclinical studies demonstrate that ASOs encapsulated in LNPs can be effective at substantially lower doses than naked ASOs [195]. Alternative delivery systems, including exosome-based and synthetic polymer-based carriers, have also shown encouraging results in improving ASO delivery in target tissues [196,197,198].

While improved delivery is critical, optimizing ASO sequence and chemical modifications remain equally essential. The following section provides a comprehensive overview of emerging AI-based platforms for designing ASO sequences and modification patterns to maximize efficacy.

## 4. Platforms Utilizing AI to Optimize ASO Design

One approach to overcoming the limitations of suboptimal ASO efficacy is the development of platforms aimed at using AI to optimize ASO design. There are two current platforms that exist, eSkipFinder and ASOptimizer [18,20]. These platforms generally follow a similar framework. They are trained on comprehensive databases comprising published experimental data relevant to their platform. By analyzing this data, the machine learning models learn to identify characteristics of the sequences that are strongly associated with improved efficacy. Researchers can then use these platforms to compare predicted efficacies across multiple candidate sequences or to directly identify the most promising ASO design for their target (Figure 1). ASOptimizer also offers predictions on optimal chemical modification patterns to further enhance therapeutic performance. By integrating a wide range of ASO design parameters, these tools offer a unique and effective approach to enhancing the precision and effectiveness of ASO therapeutics.

### 4.1. eSkipFinder: Optimizing Exon-Skipping ASOs

eSkipFinder (https://eskip-finder.org, accessed on 14 July 2025) is a machine learning platform specifically designed to support the development of fully chemically modified ASOs for exon-skipping applications [18]. The platform involves a support vector regressor (SVR) to rank the predicted exon-skipping efficiency of ASO sequences ranging from 15 to 30 base pairs in length in the input target exon sequence. Supporting the platform is a comprehensive database compiled from all publicly available English-language literature and patent records reporting exon-skipping ASOs. This database includes key features such as ASO sequence, chemical modifications, and target exon, along with relevant experimental conditions and observed exon-skipping efficiencies. To train and validate the prediction model, specific criteria were utilized to identify only ASOs targeting the dystrophin gene with sufficiently detailed experimental data. Specifically, those reporting the concentration used and the corresponding quantitative exon-skipping efficiency. The resulting 426 entries for 109 novel ASO sequences were then appropriately separated into training (90%) and testing (10%) groups. 32 features of the ASOs and the target sequence were assessed, including features like GC content of the ASO and the target, predicted binding score, ASO length, and mRNA target accessibility. Numerous SVR models were created, using up to 7 of these features, and the resulting R^2^ values were compared between all models tested to identify the most important features. Of the 32 features evaluated, ASO concentration, total GCs in sequence, distance from splice acceptor site, GC content of exon when blocked by the ASO, cumulative NI score per base [199], and predicted accessibility of 3′ end of the target were determined to be the features with the most significant effect on ASO efficacy. Due to the distinct differences in the effect of 2′-OME and PMO chemical structures, inputs for these ASOs were separated for training and testing. The final model received a predictive R^2^ value of 0.6 and 0.7 for PMO and 2′-OME modified ASOs, respectively. To further validate the effectiveness of the model beyond DMD, the platform was applied to PMOs targeting *COL7A1* and found to be effective in predicting the efficacy of input sequences.

A recent study evaluated an updated version of eSkipFinder, developed to improve its predictive accuracy and reduce computation time [200]. In this version, the original SVR model was replaced with an ensemble learning approach combining random forest, gradient boosting, and XGBoost algorithms. The ensemble approach outperformed the prediction power of the original SVR model, achieving an R^2^ value of 0.706 for PMOs and 0.795 for 2′-MOE-modified ASOs. The platform also generated predictions in under 10 s.

### 4.2. ASOptimizer: Optimizing RNase H-Dependent ASOs

On the other hand, ASOptimizer (https://asoptimizer.s-core.ai/, accessed on 14 July 2025) was developed for the purpose of optimizing the sequence and chemical modification patterns of ASOs, up to 22 base pairs in length, utilizing RNase H-dependent mechanisms to reduce target RNA expression [20]. Their database, consisting of 187,090 entries targeting 67 unique mRNAs, was created by collecting all experimental in vitro and in vivo data on ASOs targeting mRNA for degradation in published literature and patents. Similar to eSkipFinder, the comprehensive entries included all the features of the experiment, ASO sequence, and effect on target RNA. Uniquely, as a range of chemical modification patterns can be utilized to initiate RNase-H dependent mechanisms, the chemical modifications and the pattern of them used throughout the ASO sequence are also included. In total, 187,090 entries for 67 unique mRNA targets made up the database.

The first step of the platform is a linear factor model to identify sequences for the specified target with the highest predicted efficacy based on the sequence’s thermodynamic properties and predicted secondary structural features of the target mRNA. In comparison to the more traditional ASO and siRNA sequence design methods, *Sfold*, which achieved a Pearson correlation of 0.50 when predicting ASO efficacy for *IDO1* inhibition, ASOptimizer exhibited a stronger predictive capability with a Pearson correlation of 0.66 when trained and tested on *IDO1* data. For longer sequences, ASOptimizer was able to achieve a *ρ* = 0.72. To determine this model’s generalizability, they trained the model on experimental data for *PLP1* and *APOL1* and then tested the model on data for *IDO1*. Achieving only a modest decrease in score compared to when the model was trained on the target’s experimental data, the model still achieved a Pearson correlation of 0.60. The six top 19-base pair ASO sequences targeting IDO1, identified using the linear factor model of the ASOptimizer system, were synthesized with a PS backbone and were all shown to be effective in reducing *IDO1* mRNA and protein levels in HeLa cells.

The second component of the ASOptimizer platform utilizes an edge-augmented graph transformer-based deep neural network to predict the most effective chemical engineering strategy for a given ASO. When trained on a subset of the database, the model achieved 73.39% accuracy in ranking unique chemical modification patterns by their predicted effectiveness on the remaining test pairs. To assess its performance, the six previously identified IDO1-targeting ASO sequences from phase one of ASOptimizer were advanced into the sequence engineering phase. To reduce complexity, the model was restricted to patterns incorporating LNA modifications, as these were the most extensively represented in the database. Each ASO sequence was then synthesized into two formats: a conventional gapmer design with five flanking nucleotides on each end chemically modified, and the optimized design generated by ASOptimizer. Interestingly, the optimized designs often diverged from standard gapmer conventions, frequently having fewer modified nucleotides at the flanks or introducing unmodified nucleotides within the modified region (Figure 2). Notably, ASOs incorporating ASOptimizer-identified chemical patterns demonstrated enhanced activity in reducing *IDO1* expression, with several significantly outperforming their conventional gapmer counterparts.

### 4.3. Application of ASOptimizer and eSkipFinder for the Design of ASOs

Since becoming publicly accessible, eSkipFinder has successfully been used by multiple groups to support ASO design (Table 2). For example, Anwar et al. (2024) used eSkipFinder to design PMOs targeting exon 27 of the *DYSF* gene, aiming to restore the reading frame disrupted by mutations that cause exon 26 skipping, which results in dysferlinopathy (OMIM# 253601) [201]. The designed sequence achieved a 65–92% exon exclusion in patient-derived cells. To date, this remains the only published application of eSkipFinder for ASO development in rare neuromuscular disorders. Beyond this, eSkipFinder has only demonstrated utility in designing ASOs to exclude poison exons in *SCN1A* transcripts for Dravet syndrome [57], as well as exon 3 of *KRAS* transcripts for the treatment of lung adenocarcinoma [202].

As these platforms have only recently become available, the studies reporting their use have all been published within the last year. It is likely that additional research highlighting the application of eSkipFinder and ASOptimizer in ASO design will emerge in the coming years.

### 4.4. Future Directions for Machine-Learning Platforms Optimizing ASO Design

AI-driven platforms advanced ASO sequence and chemical modification design beyond traditional in silico programs, which are limited to optimizing only one or two features at a time [19,20]. By enhancing ASO efficacy at the design stage, these platforms can reduce reliance on labor- and time-intensive in vitro studies. Moreover, several FDA-approved ASOs may not represent the most optimal sequences for their intended targets, highlighting the need for improved design strategies [17]. Integrating AI into this process, moving beyond traditional methods that rely primarily on empirical observations or computational tools with limited capacity to account for ASO complexity [203], offers the potential to identify more effective candidates earlier. This, in turn, would be expected to accelerate the development of highly efficient therapies and improve outcomes for patients.

Despite their promise, these platforms have important limitations that warrant further consideration, most notably their lack of emphasis on safety during the design process. Both ASO sequence and chemical modifications are critical determinants of safety as the sequence influences the potential for off-target effects, while chemical modifications can trigger toxicity [12,175]. Because current platforms give minimal attention to these factors, it remains essential that ASOs designed through them undergo rigorous evaluation for toxicity and off-target activity. Incorporating safety parameters directly into the design stage would not only improve confidence in the resulting sequences and modification patterns but also facilitate the development of safer, more clinically viable ASOs. To maximize effectiveness, it will be important to train these platforms using experimental data from the liver and kidney, focusing on key markers of cytotoxicity and nephrotoxicity, as these organs are known to accumulate the drug upon systemic injection [204].

In addition, the generalizability of these platforms still warrants further validation. Currently, the ability of eSkipFinder to predict ASO efficacy based on sequence has only been assessed using previously published data for *COL7A1* and lacks direct in vitro or in vivo validation [18]. ASOptimizer has received additional support through in vitro analyses demonstrating effective *IDO1* downregulation following both sequence and chemical optimization [20]. However, the current evidence is limited to a single gene target, and both platforms would benefit from more comprehensive validation by comparing optimized ASOs with traditionally designed counterparts in both in vitro and in vivo models. Additionally, it will be important to assess their performance across a broader range of gene targets to confirm their versatility [198].

As additional ASO experimental data becomes available and integrated into their training datasets, the predictive accuracy of these platforms is expected to improve, enhancing their ability to design effective ASOs for diverse therapeutic applications in rare neuromuscular disorders [205]. Specifically, expanding the experimental datasets for both platforms to include a wider range of targets, a greater number of ASOs, and more diverse chemical modifications would substantially improve their generalizability and predictive accuracy, particularly for less-studied mRNA targets. ASOptimizer would benefit from expanding its focus beyond LNA-based chemical modification patterns, given their known safety limitations [206]. Furthermore, incorporating additional negative findings into training data would enhance model performance by better distinguishing between effective and ineffective ASO designs. As both tools currently rely primarily on experimental data from English-language patents and publications, expanding their datasets to include information from patents and studies originating in other countries would greatly enhance their power.

As these platforms gain wider adoption among researchers developing therapeutics for rare neuromuscular diseases, several limitations should be considered, particularly to support investigators with varying levels of experience in ASO development. A key limitation is that current platforms are not yet applicable to exon-inclusion. Optimizing ASOs for exon inclusion is especially challenging due to the limited amount of experimental data available compared with other ASO mechanisms, and the development of AI-driven tools in this area could significantly benefit the field [68]. Another important consideration is the role of delivery systems and molecular conjugates, which hold significant potential to enhance therapeutic efficacy [155,159]. In the future, it would be highly advantageous for these platforms to incorporate predictive capabilities for delivery methods or conjugates best suited to specific target tissues and chemical modifications applied, thereby maximizing overall therapeutic effectiveness.

## 5. Conclusions

AI-based ASO design platforms hold considerable promise for advancing the clinical translation of ASOs in rare neuromuscular diseases, while also making effective ASO design more accessible to researchers outside the field [18,20,203]. Their widespread adoption remains uncertain, as these tools have only recently become available and publications demonstrating their application are still limited [57,201,202]. Nonetheless, limitations in these platforms still exist. Current platforms lack comprehensive in vitro and in vivo validation of optimized ASOs relative to conventionally designed counterparts; their generalizability is uncertain due to the small number of genes evaluated, and safety considerations are often overlooked. Addressing these limitations, particularly through experimental validation and continual integration of new datasets, will be critical to realizing their full potential. Through iterative refinement, these platforms are well-positioned to evolve alongside ASO research and substantially enhance the development of effective therapeutics.

## Figures and Tables

**Figure 1 genes-16-01468-f001:**
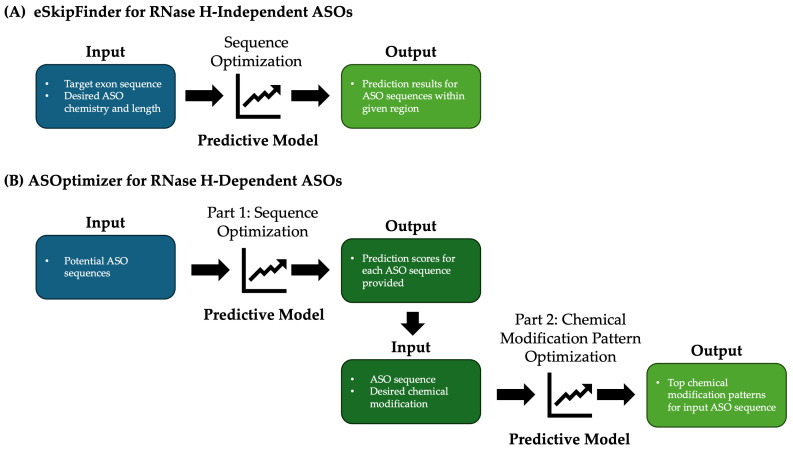
Overview of AI-based ASO optimization platforms. (**A**) eSkipFinder, designed for RNase H-independent ASOs, allows users to input a target exon sequence along with the desired ASO chemistry and length. The platform then uses a predictive model to generate predicted efficacy scores for candidate ASO sequences within the specified region. (**B**) ASOptimizer, developed for RNase H-dependent ASOs, operates in two stages. The first predictive model evaluates input ASO sequences to generate efficacy scores, while the second model identifies optimal chemical modification patterns for the selected ASOs.

**Figure 2 genes-16-01468-f002:**
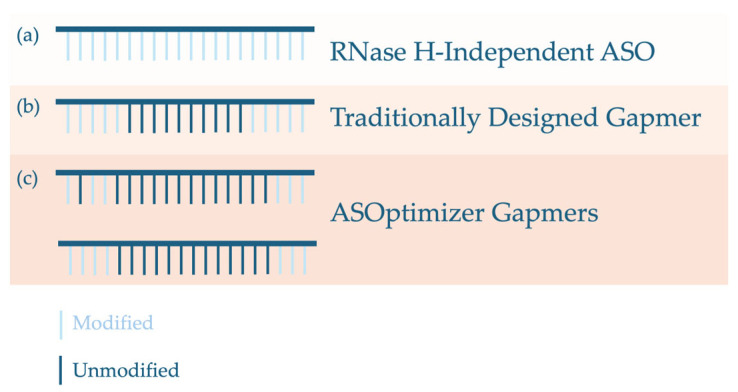
Chemical patterns generated from ASOptimizer often deviate from conventional gapmer design. (**a**) RNase H-independent ASOs are typically fully chemically modified across all nucleotides. (**b**) Traditional gapmers are designed with chemical modifications restricted to the five flanking nucleotides at both ends of the ASO. (**c**) In contrast, ASOptimizer-derived gapmers frequently depart from these canonical patterns, incorporating unmodified nucleotides within flanking regions or utilizing fewer than five chemically modified nucleotides per side.

**Table 1 genes-16-01468-t001:** Summary of FDA-approved ASOs, including year of approval, target disease, target gene (and exon in brackets, where applicable) mechanism of action, chemical modifications, conjugation strategies, and standard delivery methods.

Drug Name	Year Approved	Disease	Mechanism	Target Gene/Exon	Modifications	Conjugations	Delivery Method
Eteplirsen	2016	DMD	Exon skipping	*DMD* (51)	PMO	-	Intravenous
Nusinersen	2016	SMA	Exon inclusion	*SMN2* (7)	PS; 2′-MOE	-	Intrathecal
Inotersen	2018	hATTR	RNA degradation	*TTR*	PS; 2′-MOE	-	Subcutaneous
Golodirsen	2019	DMD	Exon skipping	*DMD* (53)	PMO	-	Intravenous
Viltolarsen	2020	DMD	Exon skipping	*DMD* (53)	PMO	-	Intravenous
Casimersen	2021	DMD	Exon skipping	*DMD* (45)	PMO	-	Intravenous
Eplontersen	2023	hATTR	RNA degradation	*TTR*	PS; 2′-MOE	GalNAc	Subcutaneous
Tofersen	2023	ALS	RNA degradation	*SOD1*	2′-MOE	-	Intrathecal

DMD, Duchenne muscular dystrophy; SMA, spinal muscular atrophy; hATTR, hereditary transthyretin amyloidosis; ALS, amyotrophic lateral sclerosis.

**Table 2 genes-16-01468-t002:** Overview of applications of AI-based platforms for ASO design, showing the target gene and exon, along with the year of publication.

Disease	Target Gene	Target Exon	Year
Dysferlinopathy	*DYSF*	27	2025 [201]
Dravet Syndrome	*SCN1A*	Poison Exon	2025 [57]
Lung Adenocarcinoma	*KRAS*	3	2025 [202]

## Data Availability

No new data were created or analyzed in this study. Data sharing is not applicable to this article.

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
