# Peer review of "Artificial Intelligence-Driven Design of Antisense Oligonucleotides for Precision Medicine in Neuromuscular Disorders"

_genes, 2025, doi:10.3390/genes16121468_

Round 1
Reviewer 1 Report
Comments and Suggestions for Authors
The authors of the review manuscript entitled "Artificial Intelligence-Driven Design of Antisense Oligonucleotides for Precision Medicine in Neuromuscular Disorders" present strategies for the design and optimisation of antisense oligonucleotides in the context of neuromuscular diseases, together with an overview of currently approved and investigational drugs.
I would like the authors to address the following points:
1. Tables appear to be incorrectly numbered, Table 2 is used twice (pages 7 and 13).
1. For completeness, please link each of the neuromuscular disorders highlighted to the corresponding OMIM ID/entry.
2. To improve readability, add the gene/target exon in the description of ASO drugs in Table 2 (page 7)
3. Correct the typo (capitalisation) in "fomivirsen" (page 7, line 291).
4. There appears to be a potential typo in "AI-driven tolls" (page 14, line 646).
4. Including both the predicted efficacy of exon-skipping ASOs and the observed efficacy would be highly relevant for the audience (table 2, page-13).
5. Discuss the possibility of extending ASO therapy to non-monogenic neuromuscular disorders, e.g., Myasthenia Gravis (MIM:254200).
6. Which additional types of data would be most valuable for refactoring or re-training the current tools (eSkipFinder and ASOptimizer) to enhance ASO efficacy predictions for exon skipping?
7. Do the authors anticipate any future public release of eSkipFinder in an open repository, as has been the case for ASOptimizer?
Author Response
RESPONSE TO REVIEWERS:
We appreciate the reviewers’ valuable comments and suggestions. The manuscript has been revised according to the reviewers’ suggestions, and we believe that the improvements have resulted in a stronger paper. Described below are the point-by-point responses to the comments and suggestions provided by the reviewers and the corresponding revisions made to address these comments.
RESPONSE TO REVIEWER ONE:
The authors of the review manuscript entitled "Artificial Intelligence-Driven Design of Antisense Oligonucleotides for Precision Medicine in Neuromuscular Disorders" present strategies for the design and optimisation of antisense oligonucleotides in the context of neuromuscular diseases, together with an overview of currently approved and investigational drugs.
I would like the authors to address the following points:
- Tables appear to be incorrectly numbered, Table 2 is used twice (pages 7 and 13).
Response: Thank you for acknowledging the improperly numbered tables. The caption for the table on page 7 has been updated to “Table 1.”
For completeness, please link each of the neuromuscular disorders highlighted to the corresponding OMIM ID/entry.
Response: The OMIM ID for each of the neuromuscular disorders highlighted has been included for completeness.
To improve readability, add the gene/target exon in the description of ASO drugs in Table 2 (page 7)
Response: Thank you for your thoughtful comment to include the gene/target exon to improve the completeness of Table 2. A new column has been added to include the target gene/exon and the table caption has been updated to read “Summary of FDA-approved ASOs, including year of approval, target disease, target gene (and exon in brackets, where applicable) mechanism of action, chemical modifications, conjugation strategies, and standard delivery methods.”
Correct the typo (capitalisation) in "fomivirsen" (page 7, line 291).
Response: The typo has been resolved and “Fomiversen” has been capitalized.
There appears to be a potential typo in "AI-driven tolls" (page 14, line 646).
Response: We appreciate your acknowledgment of the typo. “Tolls” has been updated to “tools” on line 646.
Including both the predicted efficacy of exon-skipping ASOs and the observed efficacy would be highly relevant for the audience (table 2, page-13).
Response: Thank you for the suggestion to include both the predicted efficacy and observed efficacy of ASOs in table 2. However, predicted efficacy scores are not provided in the publications that use eSkipFinder for ASO sequence optimization. Additionally, the observed efficacy of listed ASOs in table 2 varies substantially depending on factors such as dosing, cell type, and transfection conditions. Given this variability, including observed efficacy values in the table may be misleading or overly complex.
Discuss the possibility of extending ASO therapy to non-monogenic neuromuscular disorders, e.g., Myasthenia Gravis (MIM:254200).
Response: We agree that it would be valuable for the manuscript to include a discussion on the therapeutic potential of ASOs for non-monogenic neuromuscular diseases. As such, we have included a paragraph on lines 287-298 on page 6 that reads “Although the current FDA-approved ASOs for rare neuromuscular diseases treat monogenic diseases, there is significant potential for ASOs to also be applied to non-monogenic diseases by regulating the expression of transcripts that contribute to disease pathology. For example, myasthenia gravis (MG; OMIM #254200) is an autoimmune disease typically caused by autoantibodies targeting the acetylcholine receptors (AChRs), resulting in impaired neuromuscular transmission [123]. Patients with MG exhibit elevalted levels of the “read-through” acetylcholinesterase isoform (AChE-R), which degrades acetylcholine, a neurotransmitter essential for neuromuscular function [124,125]. ASOs targeting AChE-R to promote its degradation have shown efficacy in improving MG symptoms in early clinical trials [126,127]. While long-term studies are still needed to confirm safety and persisting clinical benefits, these findings demonstrate the broader therapeutic potential of ASOs for treating non-monogenic neuromuscular disorders.”
Which additional types of data would be most valuable for refactoring or re-training the current tools (eSkipFinder and ASOptimizer) to enhance ASO efficacy predictions for exon skipping?
Response: Thank you for the relevant question regarding the types of data that would be most valuable to enhance both tool’s predictions. The following on lines 688 to 693 on page 15 was included that reads “Expanding the experimental datasets for both platforms to include a wider range of targets, a greater number of ASOs, and more diverse chemical modifications would substantially improve their generalizability and predictive accuracy, particularly for less-studied mRNA targets. Furthermore, incorporating additional negative findings into training data would enhance model performance by better distinguishing between effective and ineffective ASO designs.”
Do the authors anticipate any future public release of eSkipFinder in an open repository, as has been the case for ASOptimizer?
Response: It is unknown at this time if eSkipFinder will have a public release in an open repository. However, the platform is currently freely accessible online. For completeness, line 524 on page 11 has been updated to include the link to the online platform for eSkipFinder: “https://eskip-finder.org”. In addition, on line 569 on page 12, the link for the online platform for ASOptimizer “https://asoptimizer.s-core.ai/” was included.
Reviewer 2 Report
Comments and Suggestions for Authors
This is a very well written manuscript. Please find below my minor comments for the authors to address.
- Please include more figures from the seminal publications for eSkipFinder and ASOptimizer to describe the workflow and compare some of the data with the optimized designs vs the unoptimized designs (traditional design). Especially since this publication is focused on AI tools.
- Pleas comment on why SVR was an appropriate machine learning model utilized by eSkipFinder compared to other ML models (Random forest etc.)
- Please comment on whether there are any length limitations using these both AI tools. Please include your answer within the manuscript.
- Do both of tools consider the secondary structure of the pre-mRNA or mRNA for assessing ASO accessibility?
- Another limitation is that these AI tools only incorporate patents and publications written in English. Sometime, there are patents from other countries which are not in English and that would be another point for the future outlook to incorporate that.
- Why was specifically LNA modifications included compared to other modifications – especially since they show a non-favorable safety profile compared to other modifications?
- As the authors point out that there is lack of safety data being incorporated into the tools. Specifically, these safety data should be from liver or kidney cells in addition to general cytotoxicity and nephrotoxicity data, since the major accumulation organs for ASO are liver and kidney.
Author Response
RESPONSE TO REVIEWER TWO:
This is a very well written manuscript. Please find below my minor comments for the authors to address.
- Please include more figures from the seminal publications for eSkipFinder and ASOptimizer to describe the workflow and compare some of the data with the optimized designs vs the unoptimized designs (traditional design). Especially since this publication is focused on AI tools.
Response: That is a valuable suggestion to include more figures describing the workflow of eSkipFinder and ASOptimizer. I have included Figure 1 in the publication providing an overview of both platforms. At present, there are no available in vitro or in vivo data directly comparing ASO sequences optimized through these platforms to those developed using traditional methods. However, the authors did evaluate ASOs incorporating ASOptimizer-identified chemical modification patterns against conventional gapmers. As stated in lines 618–621 (pages 13–14): “Notably, ASOs incorporating ASOptimizer-identified chemical patterns demonstrated enhanced activity in reducing IDO1 expression, with several significantly outperforming their conventional gapmer counterparts.”
- Pleas comment on why SVR was an appropriate machine learning model utilized by eSkipFinder compared to other ML models (Random forest etc.)
Response: Thank you for your thoughtful suggestion to include a comment on why SVR was appropriate machine learning model for eSkipFinder. A study was conducted to evaluate an updated eSkipFinder platform involving a three-way-voting approach that utilizes multiple ML models and was seen to perform better than the original SVR model. We have included a description of this on lines 572 to 578 on page 12 that reads: “A recent study evaluated an updated version of eSkipFinder, developed to improve its predictive accuracy and reduce computation time [201]. In this version, the original SVR model was replaced with an ensemble learning-approach combining random forest, gradient boosting, and XGBoost algorithms. The ensemble approach outperformed the prediction power of the original SVR model, achieving an R2 value of 0.706 for PMOs and 0.795 for 2’MOE-modified ASOs. The platform also generated predictions in under 10 seconds.”
- Please comment on whether there are any length limitations using these both AI tools. Please include your answer within the manuscript.
Response: We appreciate your suggestion to state if there are length limitations with both AI tools. The sentence on page 12, line 547, has been updated to “The platform involves a support vector regressor (SVR) to predict the exon-skipping efficiency of input ASO sequences ranging from 15 to 30 base pairs in length” to reflect the length limitations for eSkipFinder. In addition, to communicate the length limitations for ASOptimizer, line 581 on page 13, has been updated to read “On the other hand, ASOptimizer (https://asoptimizer.s-core.ai/) was developed for the purpose of optimizing the sequence and chemical modification patterns of ASOs, up tp 22 base pairs in length, utilizing RNAse H-dependent mechanisms to reduce target RNA expression”.
- Do both of tools consider the secondary structure of the pre-mRNA or mRNA for assessing ASO accessibility?
Response: Both tools do consider the secondary structure of the mRNA for ASO accessibility. Line 559 of page 12 has been updated from “and target accessibility” to “and mRNA target accessibility” to clearly state that eSkipFinder considers mRNA target accessibility. Regarding ASOptimizer, line 593 on page 13 has been updated from “predicted structural features” to “predicted secondary structural features of the target mRNA”.
- Another limitation is that these AI tools only incorporate patents and publications written in English. Sometime, there are patents from other countries which are not in English and that would be another point for the future outlook to incorporate that.
Response: That is a valuable comment that we have included in our discussion of future directions on lines 695 to 698 that reads: “As both tools currently rely primarily on experimental data from English-language patents and publications, expanding their datasets to include information from patents and studies originating in other countries would greatly enhance their power.”
- Why was specifically LNA modifications included compared to other modifications – especially since they show a non-favorable safety profile compared to other modifications?
Response: LNA modifications were included, as well as the primary focus of the evaluation of ASOptimizer, because they were experimental dataset was composed primarily of experimental data on LNA gapmers. Lines 611 to 613 on page 13 read: “the model was restricted to patterns incorporating LNA modifications, as these were the most extensively represented in the database.” To acknowledge the non-favorable safety profile compared to other modifications, “ASOptimizer would benefit from expanding its focus beyond LNA-based chemical modification patterns, given their known safety limitations [208].” has been added on lines 691 to 693 on page 15.
- As the authors point out that there is lack of safety data being incorporated into the tools. Specifically, these safety data should be from liver or kidney cells in addition to general cytotoxicity and nephrotoxicity data, since the major accumulation organs for ASO are liver and kidney.
Response: Thank you for the thoughtful suggestion to include the specific safety data that should be incorporated into these tools. The following has been included on lines 669 to 671 on page 15 that reads: “To maximize effectiveness, it will be important to train these platforms using experimental data from the liver and kidney, focusing on key markers of cytotoxicity and nephrotoxicity, as these organs are known to accumulate the drug upon systemic injection [206].”
Round 2
Reviewer 2 Report
Comments and Suggestions for Authors
The authors have made the requested revisions. This manuscript is now suitable for publication.